# Comparative and Phylogenetic Analyses of Complete Chloroplast Genomes of *Scrophularia incisa* Complex (Scrophulariaceae)

**DOI:** 10.3390/genes13101691

**Published:** 2022-09-21

**Authors:** Ruihong Wang, Jing Gao, Jieying Feng, Zhaoping Yang, Zhechen Qi, Pan Li, Chengxin Fu

**Affiliations:** 1Zhejiang Province Key Laboratory of Plant Secondary Metabolism and Regulation, College of Life Sciences and Medicine, Zhejiang Sci-Tech University, Hangzhou 310018, China; 2CAS Key Laboratory of Plant Germplasm Enhancement and Specialty Agriculture, Wuhan Botanical Garden, The Innovative Academy of Seed Design, Chinese Academy of Sciences, Wuhan 430074, China; 3Key Laboratory of Biological Resources and Conservation and Application, College of Life 9 Sciences, Tarim University, Alaer 843300, China; 4Laboratory of Systematic & Evolutionary Botany and Biodiversity, College of Life Sciences, Zhejiang University, Hangzhou 310058, China

**Keywords:** *Scrophularia*, desert and steppe subshrubs, chloroplast genome, comparative genomics, phylogenomics

## Abstract

The *Scrophularia incisa* complex is a group of closely related desert and steppe subshrubs that includes *S. incisa*, *S. kiriloviana* and *S. dentata*, which are the only *S.* sect. *Caninae* components found in Northwest China. Based on earlier molecular evidence, the species boundaries and phylogenetic relationships within this complex remain poorly resolved. Here, we characterized seven complete chloroplast genomes encompassing the representatives of the three taxa in the complex and one closely related species, *S. integrifolia*, as well as three other species of *Scrophularia*. Comparative genomic analyses indicated that the genomic structure, gene order and content were highly conserved among these eleven plastomes. Highly variable plastid regions and simple sequence repeats (SSRs) were identified. The robust and consistent phylogenetic relationships of the *S. incisa* complex were firstly constructed based on a total of 26 plastid genomes from Scrophulariaceae. Within the monophyletic complex, a *S. kiriloviana* individual from Pamirs Plateau was identified as the earliest diverging clade, followed by *S. dentata* from Tibet, while the remaining individuals of *S. kiriloviana* from the Tianshan Mountains and *S. incisa* from Qinghai–Gansu were clustered into sister clades. Our results evidently demonstrate the capability of plastid genomes to improve phylogenetic resolution and species delimitation, particularly among closely related species, and will promote the understanding of plastome evolution in *Scrophularia*.

## 1. Introduction

The figwort genus *Scrophularia* L. (Scrophulariaceae, Lamiales) comprises 200–300 species worldwide, is a taxonomically challenging and complicated lineage with great morphological diversity and high levels of reticulation evolution [1,2,3,4,5]. The genus is widespread across the broad temperate climate zone in the Northern Hemisphere, forming four diversity centers in Southwest Asia, Central Asia, the Himalayas and the Iberian Peninsula, with Iran and Turkey having the highest species richness with 49 and 60 species, respectively. Transcaucasia and Central Asia (74 species), East Asia and the Himalayas (55 species), as well as the Iberian Peninsula in Europe and adjacent regions (23 species) are three other secondary diversity centers [5,6,7]. *Scrophularia* was formally divided into two morphologically distinct sections in Stiefelhagen’s (1910) [8] widely acknowledged taxonomic treatment of the genus: *Scrophularia* sect. *Scrophularia* is characterized by perennial herbs with reticulate leaf veins, and *Scrophularia* sect. *Caninae* is characterized by perennial subshrubs and xerophytes without reticulate leaf veins. The latter primarily distributed in Central Asia and Northwest China [9]. Over the last decade, great advances and extensive explorations have been made in understanding and interpreting the reticulate phylogeny and evolution of *Scrophularia* [5,10,11,12,13,14,15]. The most comprehensively sampled phylogeny of *Scrophularia* reported to date was based on nuclear ribosomal ITS and two plastid DNA loci (*trnQ*–*rps16*, *trnL*–*trnF*) comprising 147 species [5]. However, many subclades and infrageneric relationships (e.g., species in *S.* sect. *Caninae*) were only weakly supported, and incongruences between cytoplasmic and nuclear data were discovered [5].

*S. incisa* complex (*S.* sect. *Caninae*) is a group of subshrubby plants that are primarily (but not exclusively) distributed in the deserts and steppes of Northwest China. It consists of three closely related species: *S. incisa* Weinm., *S. kiriloviana* Schischk and *S. dentata* Royle ex Benth [16,17]. They are widely used for medicinal purposes to treat fever and exanthema in Chinese Traditional Tibetan medicine (TTM) and Traditional Mongolian medicine (TMM) due to their secondary metabolites such as iridoids and phenylpropanoids [18,19]. Of these species, *S. incisa* displays a belt-like pattern of distribution, primarily in the northern Chinese provinces of Qinghai and Gansu as well as in Inner Mongolia, spreading to Central Asia in the west and to Siberia in the east [7,16]; *S. dentata* occurs in western and southern Tibet, northwest India and Pakistan; *S. kiriloviana* is found in Xinjiang Province and Central Asia [7]. However, the interspecific taxonomy is difficult due to similar morphological characters and habitat preferences (e.g., floodplains, grasslands, mountain valleys) [20], with primary differences being the calyx membrane and leaf shape, which are considered to be greatly influenced by environmental factors (Figure 1). Our preliminary phylogenetic analyses of the genus *Scrophularia* based on nrITS and cpDNA placed this species complex as a monophyletic group embedded within the *S.* sect. *Caninae* clade [21]. Population-level phylogeographic analyses focusing on the *S. incisa* complex based on plastid DNA (*trnL*-*trnF*, *psbA*-*trnH*, *trnQ*-*rps16*) indicated that the evolutionary relationships among the major clades of the *S. incisa* complex were poorly resolved with insufficient genetic informative sites [17,21]. Therefore, more efficient techniques were required to facilitate species phylogenetic identification of the *S. incisa* complex.

Plastid genomes with a relatively small size, and a conserved structure and gene content have greatly facilitated the study of comparative genomics and species evolution [22,23,24]. Compared to traditional plastid DNA markers, the complete plastid genome arrays can provide substantially more genetic variation information and significantly improve the phylogenetic resolution of the species complex [24,25,26]. This approach is highly effective and recommended at low taxonomic levels, especially for closely related species, as demonstrated in previous reports [27,28,29]. Hence, in the present study, plastomes of individuals representing all the major clades in the *S. incisa* complex were reported, specifically aiming to: (1) characterize the entire plastid genomes of the *S. incisa* complex and comparative analyses with other plastomes of the genus using publicly available data; (2) gain insights into the *Scrophularia* plastid genome’s evolutionary pattern and resolve phylogenetic relationships within the *S. incisa* complex; (3) examine variations in simple sequence repeats (SSRs), repeat sequences and mutational hotspot regions for future studies of species identification and phylogeographic studies of *Scrophularia*.

## 2. Materials and Methods

### 2.1. Plant Samples, Sequencing and Assembly

Leaves of six individuals of major clades of *S. incisa* complex, including three individuals of *S. kiriloviana* from Xinjiang, two individuals of *S. incisa* from Qinghai and one individual of *S. dentata* from Tibet, together with a closely related species *S. integrifolia* from Tajikistan, were collected (Figure 1, Table 1). The voucher specimens were deposited in herbarium of Zhejiang Sci-Tech University (HZSTU), and specimen numbers are listed in Appendix A. Genomic DNA was extracted from silica gel dried leaf material using DNA Plantzol (Invitrogen, Carlsbad, CA, USA) and sequenced on the platform Illumina Hiseq2500 at Beijing Genomics Institute (BGI, Shenzhen, China). Complete chloroplast DNA sequences of *S. buergeriana* (KP718626), *S. dentata* (MF861202), *S. henryi* (NC036943) and *S. takeimensis* (KP718628) were downloaded from NCBI for comparative analyses with *S. incisa* complex.

The de novo assembly and reference-guided combined method was implemented to assemble these raw reads into complete plastid genomes [30]. First, for each *Scrophularia* individual, raw reads of the paired-end sequencing data were trimmed to get rid of adapters and the reads with the Phred value of base calling quality score less than 20 using CLC Genomics Workbench v10.1.1 (CLC Bio, Aarhus, Denmark; http://www.clcbio.com (accessed on 22 June 2017)). Second, the left reads with high quality were assembled into contigs applying the CLC assembler, under all optimally controlled parameters as: minimum contig length of 200 bp, mismatch cost = 2, deletion and insertion costs = 3, length fraction = 0.9 and similarity fraction = 0.8. Third, because of both organellar and nuclear DNA contained in the sequencing data, all contigs were aligned and directed according to the reference plastome of *S. buergeriana* (KP718626) to obtain the correct ones by BLAST (http://blast.ncbi.nlm.nih.gov/ (accessed on 23 June 2008)). Finally, the filtered clean contigs were ordered and realigned with the reference chloroplast genome and complete chloroplast sequences of *Scrophularia* were generated in GENEIOUS v11.01 (http://www.geneious.com (accessed on 26 September 2017)) [23,26].

### 2.2. Chloroplast Genome Annotation, Comparison and Codon Usage Bias Analysis

Complete plastomes were annotated in the GENEIOUS v11.01 and DOGMA (Dual Organella GenoMe Annotator) database software [31,32,33,34]. The genes’ information about start/stop codon positions and exon/intron boundaries were accurately identified and artificially modified by reference to homologous genes of *S. buergeriana* (KP718626). Additionally, all the determined tRNA genes were further confirmed by the predicted corresponding structures in tRNAscan-SE v1.21 [35]. The plastid genome maps were drawn by OrganellarGenome [36]. Finally, the seven annotated plastid genomes were submitted to GenBank. The codon usage bias parameters including effective number of codons (ENC), relative synonymous codon usage (RSCU) value [37,38], as well as the GC contents respectively corresponding to positions of the first, second, third codon of objective coding sequences (CDS) and the whole plastomes were estimated with CUSP and CHIPS plugins in EMBOSS (http://vmbioinfo.toulouse.inra.fr/emboss (accessed on 15 July 2007)) and CODONW v1.4.2 (http://codonw. sourceforge.net/ (accessed on 12 May 2005)).

Chloroplast genome comparative analyses across these *Scrophularia* species mentioned above and the *S. incisa* complex plastid genomes were performed with mVISTA tool (genome.lbl.gov/vista/index.shtml (accessed on 1 July 2004)) in reference to the annotation of *S. buergeriana* [39]. For the purpose of discriminating hypervariable regions, all plastome sequences were aligned and subjected to diversity analysis using DNASP v5.1 [40] to evaluate the total number of mutations (*E_ta_*) and nucleotide diversity (*P_i_*) values of protein-coding and non-coding regions that were extracted following criterion of the aligned length larger than 200 bp and no less than one mutation site.

### 2.3. Characterization of Repeat Sequences and Simple Sequence Repeats

The program REPuter [41] was operated to recognize detailed position and size information of the repeat sequences, exactly involving direct (forward), inverted (palindromic), reverse and complement repeats. The constraint conditions for all the above repeat patterns were repeat size, gap size between repeats and sequence identity more than 30 bp, 3 bp and 90%, respectively. In addition, simple sequence repeats’ (SSRs) loci across these eleven cp genomes of *Scrophularia* were searched by running MICROSATELLITE (MISA) perl script [42], according to the repeat numbers of mono-, di-, tri-, tetra-, penta- and hexanucleotide SSRs more than 10, 5, 5, 3, 3 and 3 repeat units, respectively. Generally, SSRs have significant influence on genome rearrangement and recombination [43,44].

### 2.4. Selective Pressure Analysis

In genetics, the ratio (ω) of non-synonymous (Ka) and synonymous (Ks) substitution rates is usually employed to ascertain whether the selection pressure acting on the protein-coding gene exists or not [45,46,47]. In this study, we used the program of MAFFT v7 and DnaSP v5.1 [40,48] to extract and align the CDS region, also calculate values of Ka, Ks and Ka/Ks for each plastid gene. Genes’ evolutionary processes, namely, positive, purifying or neutral selection were identified as Ka > Ks (ω > 1), Ka < Ks, (ω < 1) or Ka = Ks (ω = 1), respectively [49].

### 2.5. Phylogenetic Analysis

Altogether 26 chloroplast genomes representing 7 genera of Scrophulariaceae (Appendix A) were directed at phylogenetic tree construction, including 13 representatives of *Scrophularia*, two of *Verbascum*, five of *Buddleja* and six individuals from four genera of tribe Myoporeae as the outgroup taxa based on previous studies [50,51,52]. These 26 plastome sequences were aligned with MAFFT v7 [48]. The maximum likelihood (ML) algorithm was performed in RAxML-HPC v8.2.20 [53,54], with a GTR + I + G substitution model ascertained under the Bayesian Information Criterion (BIC) in jModelTest v2.1.7 [55]. Bayesian inference (BI) of phylogeny was conducted in MrBayes v.3.2.5 [56,57], with the Markov chain Monte Carlo (MCMC) algorithm executed for 2 million generations and trees sampled every 500 generations, discarding the first quarter of calculated trees as burn-in and constructing a consensus tree from the remaining trees to compute posterior probabilities (PPs).

## 3. Results

### 3.1. Genome Organization and Features

Plastomes of six individuals of the *S. incisa* complex and one individual of *S. integrifolia* were sequenced using the Illumina HiSeq 2500 system with totals from 14,276,152 to 20,707,954 clean reads produced. Seven complete plastid genomes from four species of *S.* sect. *Caninae* with no gaps were obtained by means of de novo and reference-guided assembly. Integrating previously published chloroplast genomes of *S. dentata* (MF861202), *S. buergeriana* Miq. (KP718626), *S. henryi* Hemsl. (NC036943) and *S. takeimensis* Nakai (KP718628), the nucleotide sequences of the eight *S.* sect. *Caninae* chloroplast genomes were determined to be 152,088–152,688 bp, and the three *S.* sect. *Scrophularia* species ranged limitedly from 152,425 bp to 153,631 bp (Figure 2, Table 1). All of those plastomes presented a typical quadripartite structure as with other angiosperms, consisting of a pair of inverted repeats (IR) (50,908–51,248 bp) partitioned by the large single-copy (LSC, 83,531–84,454 bp) and small single-copy (SSC, 17,431–17,941 bp) regions [58,59,60]. The overall GC content among these cp genomes was practically identical (37.9–38.1%), whereas the regionalized GC content in the LSC, SSC and IR were 36.0–36.2%, 32.1–32.2% and 43.1–43.2%, respectively (Table 1).

The eleven *Scrophularia* plastomes coded the same series of 132 genes, including 80 protein-coding genes, 31 transfer RNA (tRNA) genes and 4 ribosomal RNA (rRNA) genes, thereinto 115 unique and 17 duplicated genes (Table 1 and Table 2). Among the 115 unique genes, 9 protein-coding genes and 5 tRNA genes contained one single intron, while 3 genes (*clpP*, *rps12*, and *ycf3*) had two introns. The region *ycf1* gene appeared as a pseudogene (Table 2), as has been shown in almost all plant chloroplast genomes sequenced at present, and it has high variability [23,58] because certain internal stop codons were found in the coding sequence. Furthermore, the partial duplicate of *rps19* at the LSC/IRa joined region lost the function of protein-coding due to the incomplete duplication [61,62].

Codon usages were fairly similar across these *Scrophularia* individuals. The codon numbers encoded in the eleven *Scrophularia* chloroplast genomes ranged from 50,696 to 51,210. The effective number of codon (ENC) values ranged from 55.423 (*S. dentata* Tibet ZG4) to 56.141 (*S. kiriloviana* Xinjiang AK1) (Appendix A). The most and least prevalent amino acids were leucine (approx. 9.9%) and tryptophane (approx. 1.3%), respectively (Appendix A). Except for *Met* and *Trp* amino acids encoded by a particular codon, the rest showed obvious codon usage bias and that partial codons appeared much more frequently. Among a set of synonymous codons, codons ending in A or T have a relatively high number and RSCU, while codons ending in C or G have a relatively low number and RSCU. In all codons, the total GC content was approximately 37.9%, and the GC content at the third position of the synonym codons was approximately 37.9% (Appendix A), which fully indicated that the codons, especially those at the third position, had a strong property of A/T, as well as being consistent with many previous studies [63,64].

Altogether, our newly obtained results generally correspond with other previously reported *Scrophularia* species [50,65], and the eleven *Scrophularia* cp genomes were highly conserved throughout evolution in various aspects of genome features concerning gene content, arrangement and GC content, as well as codon usage.

### 3.2. Contraction and Expansion of Inverted Repeats (IRs)

Generally, the expansion and contraction of boundary regions between the IR and SC always bring about change in plant chloroplast genomes’ length [27]. The IR/SC region boundaries were analyzed and compared across these eleven *Scrophularia* chloroplast genomes (Figure 3). The overall genomic structure was basically the same, and the slight expansion of the IRs might be caused by gene translocation. Specifically, the IR_a_ region extended 40 and 41 bp towards the *rps19* gene, resulting in the *rps19* pseudogene fragment being situated at the margin between the IR_a_ and LSC. The location of fragment *ycf1* with 877–910 bp was at the IR_b_ section as the SSC–IR_a_ borders stretched into the *ycf1* gene. For the *S. henryi* chloroplast genome, the *ycf1* fragment passed the IR_a_ to the SSC region by 1 bp. Moreover, the *ycf1* genes were entirely located in the SSC region for other individuals (Figure 3).

### 3.3. Genome Comparative Analysis and Molecular Marker Identification

The comprehensive comparison of sequence divergence was examined methodically by the mVISTA package for the eleven *Scrophularia* plastomes in reference to the annotation of *S. buergeriana*. The sequence comparison and alignment showing high levels of similarity with the sequence identity of a few regions less than 90%, suggested that the *Scrophularia* chloroplast genomes’ organization are structurally conservative (Figure 4).

In conformity to most studies [66,67], IRs displayed much less variation than polytropic LSC and SSC. Comparison of the coding genes, and non-coding and intron regions generated 129 loci (60 coding genes, 59 intergenic spacers, and 10 introns) from those individuals (Figure 5; Appendix A). The values of nucleotide variability (*Pi*) for these loci in the coding regions ranged from 0.00158 (*ycf2*) to 0.02438 (*rps15*); *rps15*, *ndhG*, *ycf1*, *ccsA*, *ndhF* and *rps11* with a high *Pi* value (>0.01) were considered as divergence hotspots. The *Pi* values of non-coding regions ranged from 0.00112 to 0.04866; in addition, 11 of the most divergent and variable regions with *Pi* values greater than 0.015 included *trnH*–*psbA*, *rpl32*–*trnL*, *ndhF*–*rpl32*, *trnL*–*rps16*, *ycf3*–*trnS*, *petA*–*psbJ*, *ndhE*–n*dhG*, *ndhC*–*ndhV*, *ccsA*–*ndhD*, *ycf4*–*cemA* and *rps4*–*trnT*. The hotspot and variable regions could be profitable in the development of genetic markers for the species phylogenetic identification analysis of *Scrophularia*.

### 3.4. Characterization of Repeat Sequences and SSRs

Altogether there were 473 repeats detected in the eleven *Scrophularia* cp genomes with REPuter program [41], including 238 forward, 223 palindromic, 8 reverse and 4 complement repeats. *S. incisa* (Qinghai XH7) comprised the maximum number of repeats (52), whereas *S. dentata* possessed the least (38) (Figure 6a). The percentage of the repeats’ size ranging between 30 and 39 bp reached 84.8% for all the individuals of *Scrophularia* (Figure 6b; Appendix A). According to the principle that the fragments of the same length discovered in homologous DNA regions were considered as shared repeats [27,28], 32 investigated repeats were shared by the eleven *Scrophularia* cp genomes. Additionally, *S. incisa* (Qinghai XH7) and *S. takesimensis* exhibited the most distinct repeats (8), while *S. henryi* and *S. buergeriana* contained the fewest number (2), respectively (Appendix A).

SSR is a repetitive sequence consisting of simple repeating units in series and has been extensively used as the effective molecular marker for phylogenetic research and genetic diversity analysis [68,69]. A total of 493 SSRs were recognized across the 11 *Scrophularia* plastomes through MISA analysis identification. The amount of SSRs per single plastome ranged between 40 (*S. integrifolia* Tajikistan H8) and 53 (*S. incisa* Qinghai XH7), with 11 SSRs shared among all plastomes (Appendix A). Among these SSRs, more than three-quarters (76.2%) were constituted by A/T base components (Figure 7a). Maybe owing to the reason that the intergenic spacers (IGS) possessed a relatively higher level of mutation rates [27,29], most SSR loci lay in IGS (62.3%), followed by CDS (24.7%) and introns (13.0%) (Figure 7b). Generally, the cp genome SSRs revealed abundant genetic variability, and therefore can be applied to the study of the population genetics of *Scrophularia* species in the future.

### 3.5. Selective Pressure Analysis

We calculated the non-synonymous (Ka) and synonymous (Ks) substitution rates together with the Ka/Ks (ω) value to detect the selective pressure on the 80 cp genes of eleven *Scrophularia* individuals. The average Ka/Ks values among the *S. incisa* complex ranged from 0.0001 (*S. dentata* Tibet ZG4) to 2.65591 (*S. incisa* Gansu DJ1), and the highest Ka/Ks value was 3.2702 (*S. buergeriana*) among these 11 *Scrophularia* individuals (Appendix A, Figure 8). In fact, positive selection for the eleven individuals was exclusively observed in *S. incisa* (Gansu DJ1), and other genes had higher synonymous than non-synonymous changes, suggesting high evolutionary constraints in the *Scrophularia* genome species. The Ka/Ks of most genes were less than 1, indicating that they may be subjected to purification options, i.e., mutations were detrimental and eliminated in the population (Appendix A). Among them, 21 groups had Ka/Ks values greater than 1: the *rpl22* gene of *S. takesimensis* (KP718628) (Ka/Ks = 1.01824, *p* > 0.05), the *rps15* gene of all the individuals expect *S. buergeriana* (KP718626) and *S. henryi* (MF861203), and the *ycf15* gene of all the individuals (average Ka/Ks = 2.1879, *p* > 0.05) (Figure 8).

### 3.6. Phylogenetic Analyses

Both ML and BI phylogenetic reconstruction analyses of the 26 complete cp genomes from Scrophulariaceae (Appendix A) generated identical tree topologies and achieved high support values in each branch. The phylogenetic tree nodes nearly received fully strong support (PP/BS =1/100), with three monophyly groups manifested involving tribe Scrophularieae, tribe Buddlejeae and tribe (Myoporeae + Leucophylleae) as outgroup taxa (Figure 8). *Buddleja* was sister to *Verbascum* and *Scrophularia*, and *S.* sect. *Caninae* was resolved as a robust monophyletic clade. Within *S.* sect. *Caninae*, *S. integrifolia* (Tajikistan H8) was sister to the clade of the *S. incisa* complex, and within the complex *S. kiriloviana*, AK1 was identified as the earliest diverging clade, followed by *S. dentata* from Tibet, while the remaining individuals of *S. kiriloviana* (ZS3,AH18) and *S. incisa* (XH7,DJ1) were clustered into sister clades (Figure 9). Here, the phylogenetic relationship pattern within the *S. incisa* complex was consistent with our previous phylogeographic study based on chloroplast DNA fragments with relatively weak support [17,21]. In conjunction with previous molecular dating of the *S. incisa* complexes’ Plio-Pleistocene divergence [17,21], we speculated that speciation and population divergence within the *S. incisa* complex may be due to the vicariance and fragmentation distribution triggered by aridification increasement and sand deserts’ enlargement, which was brought by monsoon-dominated climate patterns and geological changes in the QTP *sensu lato* in Northwest China [70,71,72,73]. We also suggested that the application of a whole cp genome as a super-barcode significantly improves the effectiveness and brings a better resolution within the closely related taxa than the traditional markers of universal or specific DNA barcodes, which provide too little variable genetic information in a certain group of species [25,58].

## 4. Discussion

### 4.1. Comparative Genomics

Our comparative results revealed that the genome structure and features including gene content, gene arrangement and GC content, as well as codon usage within both the eight *S.* sect. *Caninae* individuals and the eleven *Scrophularia* cp genomes were highly conserved, which was also consistent with the general nature of the highly conserved chloroplast genomes in angiosperms. The determined nucleotide sequences of the eight *S.* sect. *Caninae* chloroplast genomes ranged narrowly from 152,088 bp to 152,868 bp, and the three *S.* sect. *Scrophularia* species ranged from 152,425 bp to 153,631 bp (Figure 2, Table 1). The amount of protein-coding, rRNA, and tRNA genes in these individuals were identical. The overall GC contents among these plastomes were almost equal (37.9–38.1%), with 36.0–36.2%, 32.1–32.2% and 43.1–43.2% corresponding to the LSC, SSC and IR, respectively (Table 1). This conservative property of *S.* sect. *Caninae* plastid genomes was probably due to the morphological character and habitat preference similarities as well as the close affinities between them, while the cp genome conservative property of the genus *Scrophularia* was suggested due to the low chloroplast substitution rate and relatively recent evolutionary divergence time [17,21,50]. There are pseudogenes, *ycf1* and *rps19*, which deserve closer scrutiny. Comparative genomics have shown that the *ycf1* gene is found in almost all plant chloroplast genomes sequenced up to now and has high variability [74,75,76]. The IR/LSC boundaries of *Scrophularia* stretched into *rps19*, which was congruent with many standard angiosperms’ chloroplast genome feature [61]. IR contraction and expansion is a common evolutionary phenomenon [77,78,79,80,81] and may cause variation in the lengths of plastid genomes [80]. The slight differences in the IR/SC boundaries in *Scrophularia* may be caused by IR contraction/expansion. We performed full sequences’ alignment and nucleotide polymorphism analysis on 11 individuals, and the results indicated that the IR regions were more conserved than the LSC and SSC (Figure 4). Maybe due to the consensus when a mutation occurs in IRs, the repetitive gene is corrected by gene conversion, thus reducing the frequency of variation in the remodeling [82]. In addition, because of this, the IRs play a significant role in stabilizing the plastid structure.

SSRs are often used as molecular markers for population genetic studies. We identified 40 to 53 chloroplast SSRs (Appendix A) in the eleven individuals that can be applied to conduct population genetic research of *Scrophularia* in the future. We performed a selective pressures analysis on the genes, and the results showed that the Ka/Ks of most genes was fewer than 1, and they may have received purifying selection. The Ka/Ks of *rpl22* in *S. takesimensis* (KP718628) was 1.01824, but it was less than 1 in other individuals, so we thought that this group of data does not indicate that the gene has received positive selection.

### 4.2. Phylogenetic Relationships within S. incisa Complex and Scrophulariaceae

Phylogenetic analysis can ordinarily provide important insights into the organism’s origin and evolution [83,84,85]. Scrophulariaceae is now known as being composed of the eight tribes: Aptosimeae, Buddlejeae, Hemimerideae, Leucophylleae, Limoselleae, Myoporeae, Scrophularieae and Teedieae [52,86,87]. The phylogeny and relationship of the genus and tribes in Scrophulariaceae were not well resolved because of the deficiency of the evidence and genetic variation information [86,87]. Our objective phylogenetic relationships within the *S. incisa* complex were reconstructed in the framework of Scrophulariaceae based on the complete chloroplast genomes. The phylogenomic analysis of Scrophulariaceae, based on current species coverage of 26 cp genomes representing four tribes, provided full support for the monophyly of the three groups of tribe Scrophularieae, tribe Buddlejeae and tribe Myoporeae + Leucophylleae (Figure 9). Meanwhile the tree also strongly supported sister relationships of the monophyly tribe Myoporeae with *Leucophyllum frutescens* from tribe Leucophylleae, and *Buddleja* with *Verbascum* + *Scrophularia*. Monophyly of the genus *Verbascum* with respect to the genus *Scrophularia* was sufficiently supported, which was consistent with available results [5,12,14].

Our present study contributes a phylogenetic framework to the chloroplast genome evolution of Scrophulariaceae, especially for the *S. incisa* complex within genus *Scrophularia*. As we obtained the results here by the ML and BI analyses, *Scrophularia* can be divided into two monophyletic groups: *S.* sect. *Scrophularia* and *S.* sect. *Caninae* (Figure 9). Within the *S.* sect. *Caninae*, *S. integrifolia* (Tajikistan H8) from the west of the Tianshan Mountains in Tajikistan was at the basal position, followed by an *S. kiriloviana* individual (Xinjiang AK1) from the eastern foothills of the Pamirs Plateau in Southwest Xinjiang, which was consistent with our previous phylogeographic study of the *S. incisa* complex based on *psbA*-*trnH*, *trnL*-*trnF* and *trnQ*-*rps16* markers [17,21]. Within the core *S. incisa* complex, a topology of (*S. dentata* + (*S. incisa* + *S. kiriloviana*)) was strongly supported. This result solves the polytomy of the *S. incisa* complex in the previous study [17,21]. In addition, although *S. kiriloviana* from Xinjiang does not form a single monophyletic cluster, maybe due to introgression and/or incomplete lineage sorting between *S. kiriloviana* AK1 and neighboring *S. integrifolia*, the *S. incisa* complex is subdivided into three distinct gene pools and all the *S. kiriloviana* individuals show dominant red genetic clusters from the Xinjiang gene pool based on previous nSSR population data [17,21]. Therefore, we suggest that there are three recognized species in the *S. incisa* complex. In combination with our previous ancestral distribution reconstruction analysis and species distribution range of *S.* sect. *Caninae* [5,9,17,21], the robust phylogenetic evidence here firmly support what we have been suggesting about the Central Asian origin of the *S. incisa* complex with subsequent diversification on the Qinghai–Tibet Plateau (QTP), the Tianshan Mountains, the Mongolian Plateau and surrounding regions. This pattern has been explored across various genera, such as *Solms-laubachia*, *Incarvillea* and *Myricaria* [88,89,90,91]. However, multi-marker time-calibrated phylogeny based on a more comprehensive sampling and both the chloroplast and nuclear genomes should be built to investigate this biogeographic scenario more rigorously.

## 5. Conclusions

The present study was designed to improve the phylogenetic resolution within the desert and steppe subshrubs *S. incisa* complex by comparing the complete chloroplast genome sequences of the main representatives covering all the lineages within the *S. incisa* complex. This analysis revealed that their chloroplast genomes are highly conserved in terms of length, genomic structure, gene number and arrangement. Close relationships between these three taxa were inferred and confirmed depending on high chloroplast genome similarities and the topology of phylogenetic trees in which *S. incisa*, *S. kiriloviana* and *S. dentata* form a strong supported distinct cluster within the *S.* sect. *Caninae* clade of genus *Scrophularia*. Recognition of highly variable regions and distinct microsatellite loci patterns in the chloroplast genomes of the *S. incisa* complex and three other species of *Scrophularia* could potentially be used in the future for identification and discrimination of these taxa. These analyses of chloroplast genomes consequently contributed to our understanding of the plastid phylogeny of the *S. incisa* complex and provide valued implications for the evolutionary history and conservation of these medicinal plants, as well as the genus *Scrophularia* in general.

## Figures and Tables

**Figure 1 genes-13-01691-f001:**
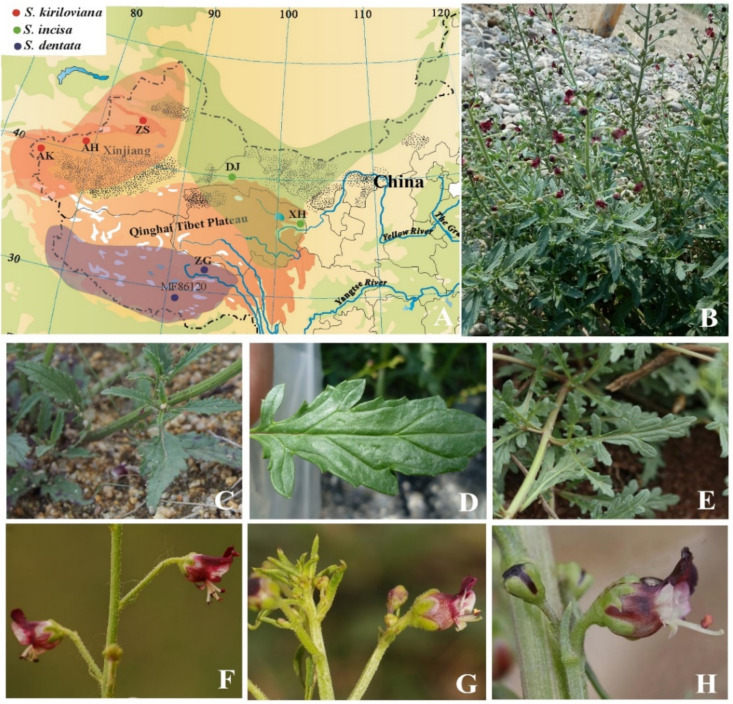
Distribution map and morphological characters of *S. incisa* complex. (**A**) Distribution map of *S. incisa* complex. (**B**) Morphology and habitat of *S. incisa*. (**C**–**E**) Leaves of *S. incisa*, *S. kiriloviana* and *S. dentata*. (**F**–**H**) Flowers of *S. incisa*, *S. kiriloviana* and *S. dentata* (Photo (**B**): by Chong-Lie Qu, (**C**): by Guo-Jun Hua, (**D**,**G**): by Pan Li, (**E**): by Xin-Xin Zhu, (**F**): Jian-Bin Pan, (**H**): by Po-Po Wu).

**Figure 2 genes-13-01691-f002:**
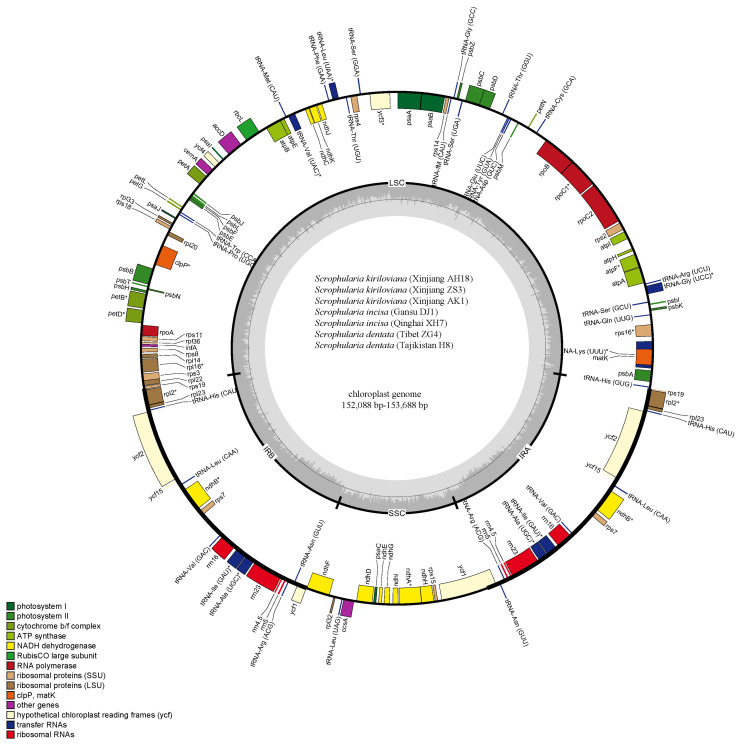
Gene maps of the seven *S.* sect. *Caninae* individuals including three *S. kiriloviana*, two *S. dentata*, two *S. incisa* and one *S. integrifolia*. Genes shown on the outside of the circle are transcribed clockwise, and genes inside are transcribed counterclockwise. Genes belonging to different functional groups are color-coded. The darker gray in the inner corresponds to GC content, and the lighter gray corresponds to AT content.

**Figure 3 genes-13-01691-f003:**
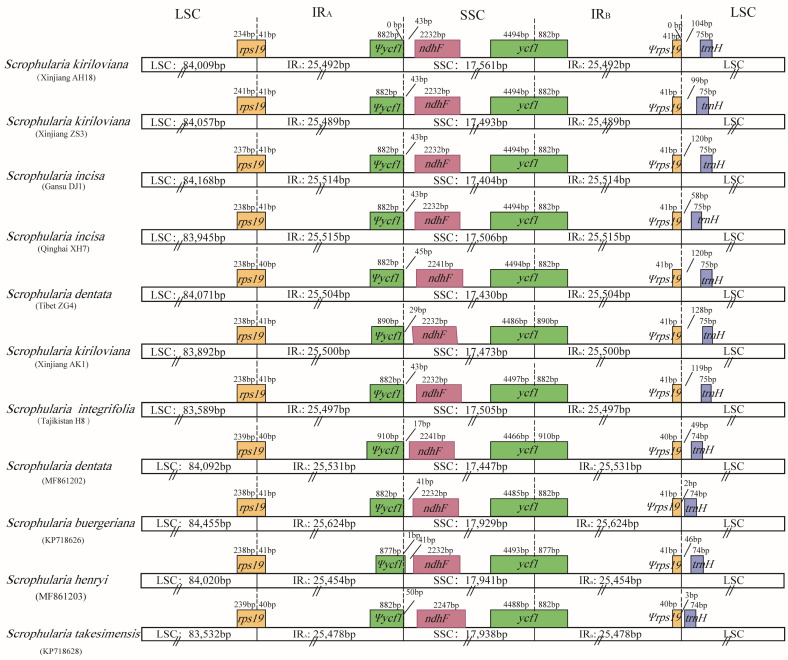
Comparison of LSC, IR and SSC junction positions among *Scrophularia* plastomes.

**Figure 4 genes-13-01691-f004:**
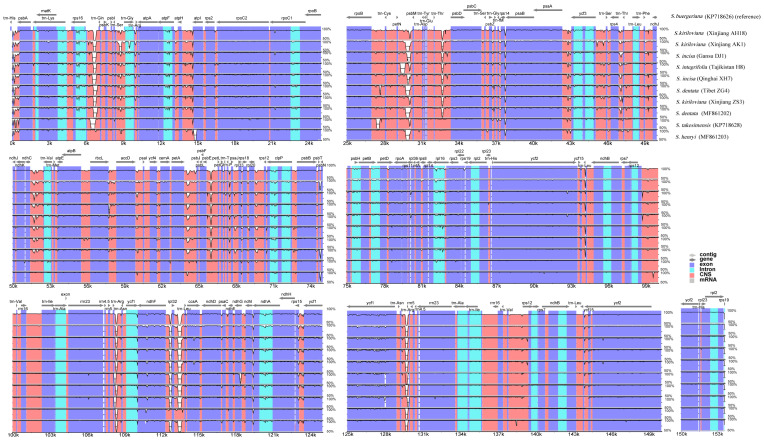
Sequence identity plots among the eleven *Scrophularia* chloroplast genomes, with *S. buergeriana* as a reference. Annotated genes are displayed along the top. The vertical scale represents the percent identity between 50 and 100%. Genome regions are color-coded as exon, intron and conserved non-coding sequences (CNS).

**Figure 5 genes-13-01691-f005:**
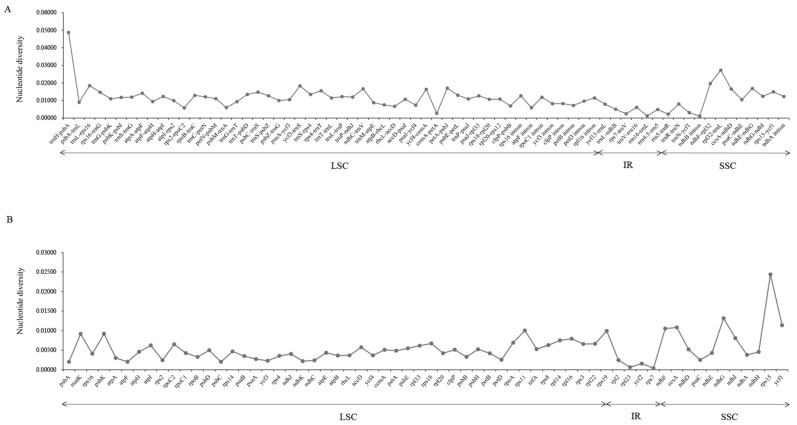
Comparison of nucleotide variability (*Pi*) values in *Scrophularia* plastomes. (**A**) *Pi* values among intergenic spacer (IGS) and intron regions. (**B**) *Pi* values among protein-coding genes (CDS).

**Figure 6 genes-13-01691-f006:**
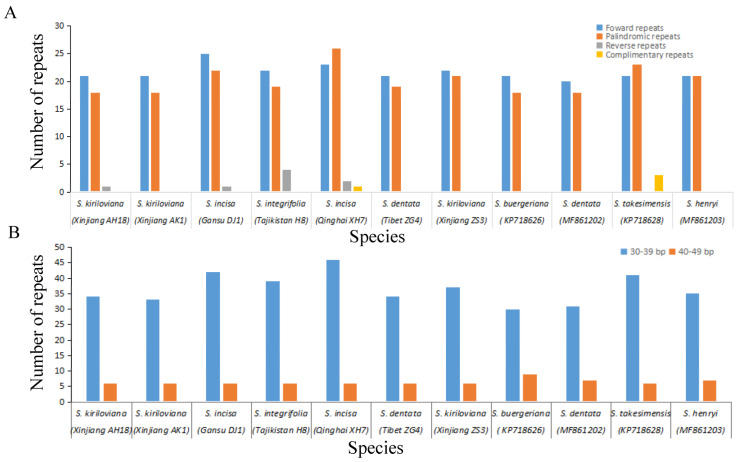
Repeat analyses in eleven chloroplast genomes of *Scrophularia*. (**A**) Frequency of repeat types. (**B**) Frequency of repeats by length.

**Figure 7 genes-13-01691-f007:**
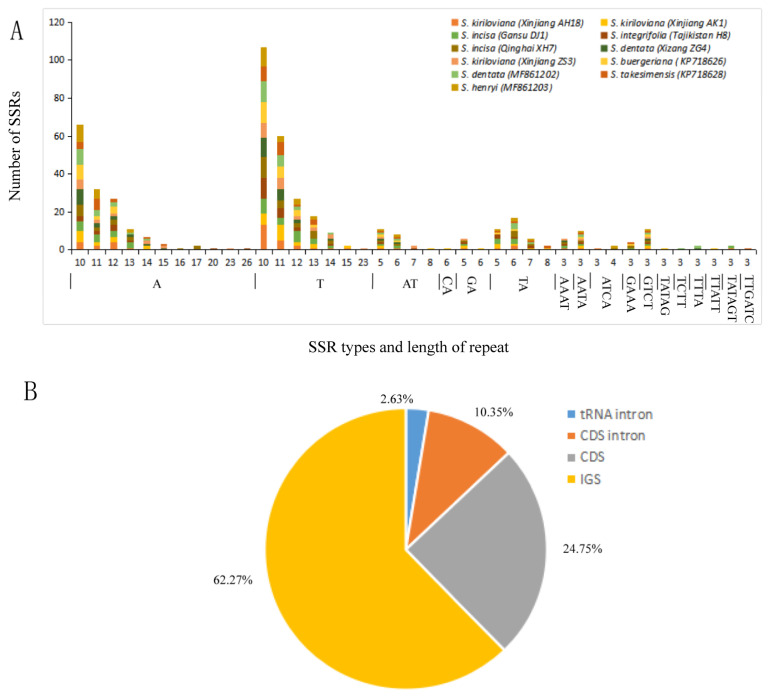
Simple sequence repeats (SSRs) in the eleven *Scrophularia* chloroplast genomes. (**A**) Number of SSRs by length. (**B**) Distribution of SSR loci. CDS, coding DNA sequence; IGS, intergenic spacer region.

**Figure 8 genes-13-01691-f008:**
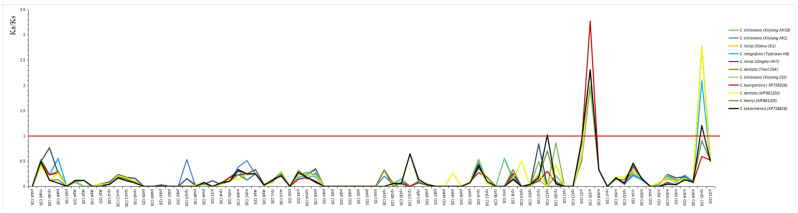
The Ka/Ks analysis of eleven individuals of *Scrophularia*.

**Figure 9 genes-13-01691-f009:**
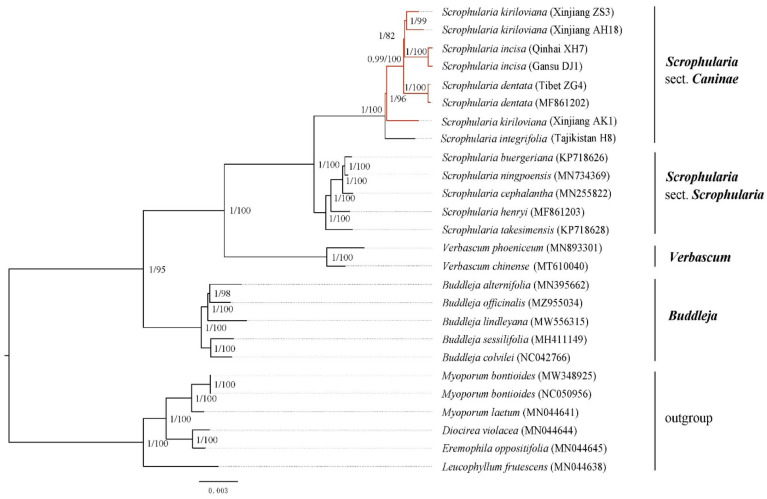
Phylogenetic relationships of *Scrophularia* species inferred from Bayesian inference (BI) and maximum likelihood (ML) based on the complete cp genome sequence dataset. Support values marked above the branches follow the order PP (posterior probability)/BS (bootstrap support).

**Table 1 genes-13-01691-t001:** The basic characteristics of *Scrophularia* chloroplast genomes.

Characteristics	*S. integrifolia*	*S. kiriloviana*	*S. kiriloviana*	*S. kiriloviana*	*S. incisa*	*S. incisa*	*S. dentata*	*S. dentata*	*S. buergeriana*	*S. takesimensis*	*S. henryi*
	(Tajikistan H8)	(Xinjiang AH18)	(Xinjiang AK1)	(Xinjiang ZS3)	(Gansu DJ1)	(Qinghai XH7)	(Tibet ZG4)				
Total cpDNA size (bp)	152,088	152,554	152,365	152,528	152,600	152,481	152,509	152,600	153,631	152,425	152,868
Latitude (°N)	36.617	40.746	38.762	43.197	39.633	35.662	29.490	30.653	——	——	31.470
Longitude (°E)	68.564	77.826	75.193	81.201	94.340	102.644	97.470	97.568	——	——	110.395
LSC length	83,589	84,009	83,892	84,057	84,168	83,945	84,071	84,091	84,454	83,531	84,020
SSC length	17,505	17,561	17,473	17,493	17,404	17,506	17,430	17,447	17,929	17,938	17,940
IR length	25,497	25,492	25,500	25,489	25,514	25,515	25,504	25,531	25,624	25,478	25,454
Total GC content (%)	38	38	38	38	37.9	38	38	38	38	38.1	38
LSC	36.1	36.1	36.1	36.1	36	36.1	36.1	36	43.2	36.2	36.1
SSC	32.1	32.1	32.1	32.2	32.1	32.1	32.2	32.2	32.2	32.2	32.2
IR	43.1	43.1	43.1	43.1	43.1	43.1	43.1	43.1	43.2	43.2	43.2
Total number of genes	132	132	132	132	132	132	132	132	132	132	132
Protein-coding genes	80	80	80	80	80	80	80	80	80	80	80
rRNA genes	4	4	4	4	4	4	4	4	4	4	4
tRNA genes	31	31	31	31	31	31	31	31	31	31	31
Duplicated genes	17	17	17	17	17	17	17	17	17	17	17
GenBank Acc. No.	OP018678	OP018676	OP036427	OP036428	OP036429	OP018675	OP018677	MF861202	KP718626	KP718628	MF861203

IR, inverted repeat region; LSC, large single-copy region; rRNA, ribosomal RNA; SSC, small single-copy region; tRNA, transfer RNA.

**Table 2 genes-13-01691-t002:** Gene composition of *Scrophularia* chloroplast genomes.

Groups of Gene	Name of Gene
Ribosomal RNAs	*rrn16*(×2), *rrn23*(×2), *rrn4.5*(×2), *rrn5*(×2)
Transfer RNAs	^a^*trnA-UGC*(×2)*, trnC-GCA, trnD-GUC, trnE-UUC, trnF-GAA, trnfM-CAU,*^a^*trnG-GCC,* *trnG-UCC, trnH-CAU, trnH-GUG, trnI-CAU,*^a^*trnI-GAU(×2), trnK-UUU, trnL-CAA*(×2)*,* ^a^*trnLUAA, trnL-UAG, trnM-CAU, trnN-GUU*(×2)*, trnP-UGG, trnQ-UUG, trnR-ACG*(×2)*,* *trnR-UCU, trnS-GCU, trnS-GGA, trnS-UGA, trnT-GGU, trnT-UGU, trnV-GAC*(×2)*,* ^a^*trnV-UAC, trnW-CCA, trnY-GUA*
Photosystem I	*psaA, psaB, psaC, psaI, psaJ*
Photosystem II	*psbA, psbB, psbC, psbD, psbE, psbF, psbH, psbI, psbJ, psbK, psbL, psbM, psbN, psbT*
Cytochrome	*petN, petA, petL, petG,* ^a^ *petB,* ^a^ *petD*
ATP synthase	*atpA,* ^a^ *atpF, atpH, atpI, atpE, atpB*
Rubisco	*rbcL*
NADH dehydrogenase	*ndhJ, ndhK, ndhC,* ^a^ *ndhB(×2), ndhF, ndhD, ndhE, ndhG, ndhI,* ^a^ *ndhA, ndhH*
ATP-dependent protease subunit P	^b^ *clpP*
Chloroplast translational initiation factor	*infA*
Chloroplast envelope membrane protein	*cemA*
Large units	*rpl33, rpl20, rpl36, rpl14,* ^a^ *rpl16,* ^a^ *rpl2(×2), rpl23(×2), rpl32*
Small units	^a^*rps16, rps2, rps14, rps4, rps18,*^b^*rps12(×2), rps11, rps8,*^Ψ^ *rps19, rps3, rps7(×2), rps15*
RNA polymerase	*rpoC2,* ^a^ *rpoC1, rpoB, rpoA*
Miscellaneous proteins	*matK, accD, ccsA*
Hypothetical proteins and conserved reading frame	^b^*ycf3, ycf4, ycf2(×2),*^Ψ^ *ycf1, ycf15(×2)*

^a^ Indicates the genes containing a single intron. ^b^ Indicates the genes containing two introns. (×2) Indicates genes duplicated in the IR regions. Pseudogene is represented by ^Ψ^.

## Data Availability

Data contained within the article are openly available in GeneBank.

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
