# Peer review of "Comparative and Phylogenetic Analyses of Complete Chloroplast Genomes of Scrophularia incisa Complex (Scrophulariaceae)"

_genes, 2022, doi:10.3390/genes13101691_

Round 1

Reviewer 1 Report

This article study to explore Comparative and Phylogenetic Analyses of Complete Chloroplast Genomes of Scrophularia incisa complex (Scrophulariaceae). This study will help to facilitate taxonomic issues among different taxons. Before recommending this article, there are some shortcomings for that should be resolve.

General comments

Overall, the study is well designed and presented in a good way, but mostly the literature is not cited and contain grammatical mistakes.  

Abstract

Abstract is well presented however main findings must be specifying e.g. results of phylogenetic study, variation among the species, respective grouping or clades according to this study.

The methods must be specified and briefly explained.

 Also add quantitative results in this section.

Introduction

The introduction part is well written but still some details are required.

Provide economic and commercial importance of the Scrophularia incisa.

The sentence should be revised “The contemporary most universally accepted taxonomic treatment”

Line 55 add reference.

Add significance of chloroplast genome and advances to study the genome of specific complexes by citing relevant study.

https://doi.org/10.3390/agronomy12051078,

Materials and methods

Add image of the studied plant in section 2.1

Section 2.5 could be cited with the following studies

 DOI: http://dx.doi.org/10.30848/PJB2022-3(19)

Discussion

Mainly the discussion is not cited.

The authors are directed to cite the relevant literature.

Conclusion

Conclusion and results are well presented

Author Response

Thank you very much for taking your time to revise our manuscript. We appreciate the editor and reviewers very much for their constructive comments and suggestions on our manuscript entitled " Comparative and Phylogenetic Analyses of Complete Chloroplast Genomes of Scrophularia incisa complex (Scrophulariaceae)". Those comments are all valuable and very helpful for revising and improving our paper, as well as the important guiding significance to our researches. According to the reviewers’ detailed suggestions, we have made a careful revision on the original manuscript. The main corrections in the paper and the responds to the reviewer's comments are as following:

General comments

1. Overall, the study is well designed and presented in a good way, but mostly the literature is not cited and contain grammatical mistakes.

Respond: Thank you for your suggestion and we have added the citation of literature references and corrected the grammatical mistakes according to the suggested advices.

Abstract

2. Abstract is well presented however main findings must be specifying e.g. results of phylogenetic study, variation among the species, respective grouping or clades according to this study.

The methods must be specified and briefly explained.

Also add quantitative results in this section.

Respond: Thank you for your suggestion and I also agree that the abstract should be more detailed. But the journal required that “The abstract should be a total of about 200 words maximum” and we had to streamline our content of abstract. However, we also have already made some changes in our manuscript about this part.

Introduction The introduction part is well written but still some details are required.

3. Provide economic and commercial importance of the Scrophularia incisa.

Respond: Up to now, the application of S. incisa and S. dentata mainly concentrated on the medicinal purposes to treat fever and exanthema in Chinese Traditional Tibetan Medicine (TTM) and Traditional Mongolian Medicine (TMM), and we have talked about it in the second paragraph of the introduction.

4. The sentence should be revised “The contemporary most universally accepted taxonomic treatment”

Respond: We have deleted the word of contemporary in the text.

5. Line 55 add reference.

Respond: We have added references at line 55.

6. Add significance of chloroplast genome and advances to study the genome of specific complexes by citing relevant study. https://doi.org/10.3390/agronomy12051078

Respond: We have added sentence about significance of chloroplast genome in the text and also added citation of relevant study. Because this article “https://doi.org/10.3390/agronomy12051078” is not about the chloroplast genome, so we have added and cited it at the second paragraph of the Introduction when talking about habitat and morphological characters.

Materials and methods

7. Add image of the studied plant in section 2.1

Respond: Thank you for your suggestion and we have added image about distribution map and morphological characters comparison of S. incisa complex in Figure 1, as well as the typical morphology and habitat of S. incisa representing the complex.

8. Section 2.5 could be cited with the following studies

DOI: http://dx.doi.org/10.30848/PJB2022-3(19)

Respond: Yes, we have cited this reference in section 2.5.

Discussion

9. Mainly the discussion is not cited.

The authors are directed to cite the relevant literature.

Respond: Thank you for your suggestion and we added the citation of relevant literature.

Conclusion

10. Conclusion and results are well presented.

Respond: Thank you for all your valuable and thoughtful comments.

Reviewer 2 Report

Dear authors,

the manuscript presets relevant new phylogenomic data on a section of Scrophularia endemic to Asia. However, the text should be carefully revised by an English native speaker since most sentences need some adjustments to make them clearer to the reader. Additionally, the discussion must be improved regarding the S. sect. Caninae systematics, since you focus only on the family. Your results evidence that one of the species you sampled is not monophyletic and you do not address this problem anywhere in the manuscript. Does it contain enough genomic differences according to your results so it could maybe in the future be segregated as a new or established species by a taxonomist expert in Scrophulariaceae?

Kind regards,

Author Response

Thank you very much for taking your time to revise our manuscript. We appreciate the editor and reviewers very much for their constructive comments and suggestions on our manuscript entitled " Comparative and Phylogenetic Analyses of Complete Chloroplast Genomes of Scrophularia incisa complex (Scrophulariaceae)". Those comments are all valuable and very helpful for revising and improving our paper, as well as the important guiding significance to our researches. According to the reviewers’ detailed suggestions, we have made a careful revision on the original manuscript. The main corrections in the paper and the responds to the reviewer's comments are as following:

1. The manuscript presets relevant new phylogenomic data on a section of Scrophularia endemic to Asia. However, the text should be carefully revised by an English native speaker since most sentences need some adjustments to make them clearer to the reader.

Respond: We apologize for the poor language of our manuscript. We worked on the manuscript for a long time and the repeated addition and removal of sentences and sections obviously led to poor readability. We have now worked on both language and readability, also have involved native English speakers for language corrections. We really hope that the language level has been substantially improved.

The manuscript also has been revised according to your comments in the pdf text, except that you suggest that use the keywords ‘Lamiales’ to replace ‘closely related species’ in Commented [M5], but we think the study is just at the family level of Scrophulariaceae, so we use ‘desert and steppe subshrubs‘ here.

And about question of Commented [M29] in the revised pdf: Were they deposited within an herbarium collection? If not, please state the nature of the biological collection in which the vouchers were deposited in this university. I have indicated that voucher specimens were deposited in College of Life Science and Medicine, Zhejiang Sci-Tech University, and specimen numbers were added in Table S1.

Thank you for your suggestions again.

2. Additionally, the discussion must be improved regarding the S. sect. Caninae systematics, since you focus only on the family.

Respond: The second paragraph of the “4.2 Phylogenetic Relationships within S. incisa Complex and Scrophulariaceae” is mainly discussed regarding S. sect. Caninae, we have made some improvements in the text now.

3. Your results evidence that one of the species you sampled is not monophyletic and you do not address this problem anywhere in the manuscript. Does it contain enough genomic differences according to your results so it could maybe in the future be segregated as a new or established species by a taxonomist expert in Scrophulariaceae?

Respond: Thank you for your good suggestion and we have added discussion about the problem that S. kiriloviana is not monophyletic. Although S. kiriloviana from Xinjiang does not form a single monophyletic cluster, maybe due to introgression and/or incomplete lineage sorting between S. kiriloviana AK1 (from eastern foothills of the Pamirs Plateau in Southwest Xinjiang) and neighbouring S. integrifolia (from the west of Tianshan Mountains in Uzbekistan), whereas S. incisa complex is divided into three distinct gene pools and all the S. kiriloviana individuals show dominant red genetic clusters from Xinjiang gene pool based on previous nSSR population data. Therefore, here we suggest that there are three recognized species in S. incisa complex.

Kind regards,

Round 2

Reviewer 2 Report

Dear authors and editor,

the manuscript was greatly improved from its first version. I sugguest minor corrections highlighted in green in the manuscript.

Author Response

Thank you very much for your careful and helpful revision on our manuscript. According to the detailed suggestions, we have made careful corrections in the pdf text.

The main corrections in the paper and the responds to the reviewer's comments are as following:

1. Commented [M4]: Is it Himalays or Himalayas?

Respond: Yes, it is Himalayas here and we have corrected.

2. Commented [M11]: replace "nuclear incongruencies" by "incongruencies in nuclear data were discovered"

Respond: I am sorry for the confused expression. In fact, we want to express the meaning of incongruences between cytoplasmic and nuclear data here, so we use “incongruences between cytoplasmic and nuclear data were discovered” to replace “cytoplasmic-nuclear incongruences were discovered”.

3. Commented [M14]: Please, specify if this institution comprises an herbarium and provide the acronym of this collection according to Thiers (2022). Also, if the vouchers were not deposited at an herbarium the authors must do so before the manuscript can be accepted for publication.

Respond: Yes, our institution comprises an herbarium. We have changed the sentence into “The voucher specimens were deposited in Herbarium of Zhejiang Sci-Tech University (HZSTU) and specimen numbers were listed in Table S1.”.

4. Commented [M15]: this sentence must be moved to the discussion section.

Respond: Thank you for your suggestion. We have moved the sentence to the discussion section, also have corrected the order of the references.

Kind regards,